# Signs, Symptoms, and Morphological Features of Idiopathic Condylar Resorption in Orthodontic Patients: A Survey-Based Study

**DOI:** 10.3390/jcm11061552

**Published:** 2022-03-11

**Authors:** Akihiko Iwasa, Eiji Tanaka

**Affiliations:** Department of Orthodontics and Dentofacial Orthopedics, Tokushima University Graduate School of Biomedical Sciences, Tokushima 770-8504, Japan; iwasa.akihiko.1@tokushima-u.ac.jp

**Keywords:** idiopathic condylar resorption, orthodontic patients, orthognathic surgery, progressive condylar resorption, temporomandibular disorders

## Abstract

Background: Idiopathic condylar resorption (ICR) is an aggressive degenerative disease of the temporomandibular joint that is most frequently observed in teenage girls. However, no specific cause of ICR has been identified. To explore the specific causes of the onset and progression of ICR, we performed a survey-based study on ICR in orthodontic patients and described its subjective symptoms, clinical signs, and condylar morphological features. Methods: A total of 1735 participants were recruited from 2193 orthodontic patients. For each participant, subjective symptoms and clinical signs of temporomandibular disorders (TMDs) were evaluated through clinical examination and a questionnaire. Furthermore, three-dimensional computed tomography (CT) was performed to diagnose ICR. Results: Among the 1735 patients evaluated, ICR was present in two male and ten female patients. All 12 patients had maxillary protrusion and an anterior open bite. Four patients with ICR underwent orthodontic treatment. Based on CT findings, patients with ICR had significantly different condylar sizes and shapes from patients with TMDs alone. Conclusions: The coexistence of intrinsic and extrinsic factors, such as sex-hormone imbalance and a history of orthodontic treatment, might lead to the onset of ICR. We suggest that growing patients suspected of having ICR should undergo CT evaluation because CT findings may precede clinical symptoms and signs.

## 1. Introduction

The temporomandibular joint (TMJ) permits large relative movements between the temporal bone and mandibular condyle [1]. Within the joint, the articular surfaces of the condyle and temporal bone are covered with a thin fibrocartilaginous layer with a low coefficient of friction [2]. During the growth period, the mandibular condyle rapidly elongates toward the temporal bone [3]. This elongation is mostly dependent on appositional growth at its apex, where chondrogenitor cells in the polymorphic cell layer differentiate into chondrocytes, which are incorporated into the underlying condylar cartilaginous tissue during chondrogenesis [4]. Hence, condylar elongation is different from that occurring in other developing long bones. Similar to other synovial joints, mechanical loading is essential for the growth, development, and maintenance of TMJ tissues, including the mandibular condylar cartilage.

Idiopathic condylar resorption (ICR) is an uncommon aggressive degenerative joint disease that most frequently occurs in teenage girls during the pubertal growth spurt [5,6]. In patients with ICR, a decrease in the mandibular ramus height occurs during the growth period, resulting in a clockwise mandibular rotation and subsequent anterior open bite [7,8,9]. Growing amounts of evidence suggest that excessive or abnormal loading with subsequent microtrauma is crucial for the development of TMJ osteoarthritis (TMJ-OA). However, unlike typical TMJ-OA, ICR is a noninflammatory but aggressive form of joint degeneration, usually without pain [10]. ICR is a well-known but poorly understood disease process with a 9:1 female-to-male frequency ratio and most frequently affects individuals aged <20 years [11,12]. Although the strong predilection for occurrence during the pubertal growth spurt in teenage girls supports the theory of hormonal mediation, no specific cause of ICR has been identified [7,8,13]. 

Arnett et al. [7,8] have suggested that ICR results from dysfunctional remodeling due to (1) a decreased adaptive capacity of the articulating structures of the TMJ or (2) excessive or imbalanced mechanical stress on the TMJ structures that exceeds the normal adaptive capacity. The former is an intrinsic host factor; the latter involves extrinsic factors and includes macrotrauma, which can directly cause destruction of the TMJ components. These factors may occur alone or may be interrelated, interdependent, or coexistent. However, TMJ deterioration and resorption under similar circumstances, including the mechanical microenvironment and general host condition, may still vary among patients.

Thus, to explore the specific causes of ICR onset and progression, we performed a survey-based study of degenerative diseases of the TMJ in orthodontic patients and described the subjective symptoms, clinical signs, and morphological features of ICR. We aimed to assess the prevalence of subjective symptoms and clinical signs indicative of ICR and their correlation with the intrinsic and extrinsic factors associated with the development of ICR. 

## 2. Materials and Methods

### 2.1. Participants

A total of 1735 participants, comprising 661 male and 1074 female patients, were recruited from 2193 patients who visited the orthodontic clinic of Tokushima University Hospital between April 2011 and December 2021. The inclusion criteria for patients were age <25 years and no history of cleft palate or craniofacial syndrome. The exclusion criteria included a history of arthritis, immune disease or systemic disease, ICR, and previous orthognathic surgery. For each participant, subjective symptoms and clinical signs of temporomandibular disorders (TMDs) were evaluated through clinical examination and a questionnaire. Furthermore, if changes and abnormal shapes of the mandibular condyle were observed on preoperative panoramic radiographs, 3-dimensional (3D) computed tomography (CT) of TMJs was performed to diagnose ICR.

The Ethics Committee of Tokushima University Hospital approved the study (permit no. 3050), and informed consent was obtained from each participant or their parents (for minor patients) after a full explanation of the research purposes and procedures.

### 2.2. Malocclusion

For all participants, the type of malocclusion was determined according to the following definitions:Maxillary protrusion, malocclusion with overjet >5 mm;Mandibular prognathism, malocclusion with anterior crossbite;Posterior crossbite, malocclusion with posterior unilateral or bilateral crossbite;Open bite, malocclusion with negative overbite;Deep bite, malocclusion with overbite >5 mm;Crowding, malocclusion with malpositioning of teeth.

In participants with several types of malocclusion, all malocclusions were included during data sampling.

### 2.3. Clinical Examination

Clinical examination was performed by four experienced orthodontists. TMDs included subjective symptoms and signs such as TMJ sounds, TMJ pain, masticatory muscle tenderness, and difficulty in mouth opening. During the initial orthodontic evaluation, TMD was diagnosed according to the following diagnostic criteria for TMD (DC/TMD) [14]:TMJ sounds, including clicking and crepitus, on joint palpation during jaw opening and closing movements.TMJ pain on joint palpation from the lateral and posterior sides.Tenderness on palpation in 10 masticatory muscles including the deep and superficial masseter, anterior and posterior parts of the temporalis, posterior belly of the digastric, sternocleidomastoid, and trapezius.Maximum pain-free mouth opening was measured using calipers. Limitations in mouth opening were defined as mouth opening < 40 mm and/or mandibular lateral deviation ≥5 mm at maximum mouth opening.

Based on the findings in the clinical examination, the patients with at least one positive finding in the masticatory muscles or TMJ were categorized as having signs of TMD.

### 2.4. Questionnaire

Patients and/or their parents were asked to complete a questionnaire about the history of TMJ pain during various mandibular movements, difficulty in mouth opening, TMJ sounds, frequent headache and stiff shoulder, masticatory muscle pain, oral parafunctions, and maxillofacial trauma. Parafunctional habits include nail biting, nocturnal bruxism, daytime clenching, thumb sucking, and tongue thrusting. In addition, multiple-choice questions regarding the following topics were added to the questionnaire: relevant medical history (arthritis, immune disease, and systemic disease), detailed medical history regarding hormonal imbalances for female patients, medications currently administered, and history of orthodontic and/or orthognathic treatment.

### 2.5. Three-Dimensional CT of TMJ

As described in Section 2.1, if abnormalities of the mandibular condyles were observed on panoramic radiographs, then 3D CT of the TMJs was performed, irrespective of TMD signs and symptoms. ICR manifested clinically or on imaging as a downregulated condylar head volume, decreased ramal height, and either progressive mandibular retrusion after puberty or reduced mandibular growth before puberty. In the patients with TMD alone as TMD group, 3D CT was performed for orthodontic treatment, that is, indications of orthognathic surgery or miniscrew-assisted rapid palatal expansion. 

The following radiological measurements were analyzed as described by Kristensen et al. [15]: (1) mandibular condylar width, length, and height and (2) condylar axial angle (Figure 1 and Figure 2). Briefly, condylar width was defined as the mediolateral dimension of the condyle, length as the anteroposterior dimension of the condyle, and height as the distance from the superior condylar point to a line perpendicular to a tangent to the ramus. The condylar axial angle was defined as the angle between the condylar axis and the midsagittal reference line (Nasion–Basion). Furthermore, on the sagittal view of the mandibular ramus perpendicular to the condylar axis, the condylar neck angle was divided into categorical data for the posterior, normal, and anterior inclinations.

To examine intra-examiner reliability, repeated measurements were conducted in five patients with 10 TMJs at 2-week intervals. The intraclass correlation coefficient (ICC) was 0.984, which confirmed the reliability of the selected measurements.

### 2.6. Statistical Analysis

Statistical analyses were performed using SPSS 22.0 (SPSS Inc., Chicago, IL, USA). The normality of each condylar morphological variable was assessed using the Shapiro–Wilk test and confirmed by an evaluation of the Q-Q plot. Normally distributed data are expressed as mean and standard deviation. Non-normally distributed data are expressed as the median and were analyzed using the Mann–Whitney *U* test. Differences in the condylar width, length, height, and axial angle between the ICR and control groups were evaluated. For data with normal distribution, a general linear model analysis for repeated measures was performed to compare the two groups. Intergroup comparisons were performed using unpaired *t*-tests. Statistical significance was set at *p* < 0.05.

## 3. Results

The overall prevalence rate of TMD in orthodontic patients was 25.9%; in particular, the prevalence rate was 20.9% in male and 29.1% in female patients (Table 1). Among the various types of malocclusion, crowding, maxillary protrusion, and mandibular prognathism were observed in 32.0%, 27.1%, and 14.7% of orthodontic patients with TMDs, respectively. Similarly, 42.4%, 20.3%, and 20.1% of patients in the control group without TMD exhibited crowding, mandibular prognathism, and maxillary protrusion, respectively (Table 2). The most common symptom was clicking; self-reported and clinically assessed joint sounds were noted in 45.1% and 58.0% of orthodontic patients with TMDs, respectively (Table 3). Regarding reported history, 64.9% of orthodontic patients with TMD had a history of parafunctional habits (Table 3).

We identified a total of 12 patients with suspected ICR. The prevalence rate was 0.7% and the male-to-female ratio was 1:5 (2 male and 10 female patients) (Table 1). Table 4 summarizes the characteristics of patients with ICR. The average age at onset was 12 years. All 12 patients had maxillary protrusion with a retrognathic chin and an anterior open bite (Table 2 and Table 4); however, 2 of the 12 participants exhibited skeletal Class III malocclusion (anterior crossbite) before the onset of ICR. Four patients had undergone orthodontic treatment and one had a history of oral contraceptive use. Notably, 9 of the 12 patients had a history of TMD, and 8 patients had parafunctional habits such as resting the chin on the hands, mouth breathing, and tongue thrusting. Compared to the TMD group, the ICR group exhibited significantly smaller maximum mouth opening (Table 3).

CT measurements revealed that the average condylar width, length, and height in the ICR and TMD groups were 14.1 ± 2.0, 8.0 ± 2.8, and 12.6 ± 4.1 mm, and 18.0 ± 1.6, 8.5 ± 1.2, and 21.7 ± 3.5 mm, respectively (Table 5). There were significant differences in the condylar width (*p* < 0.001) and height (*p* < 0.001) between the ICR and TMD groups, whereas the condylar length was not significantly different. The axial angle was 44.6° ± 12.1° in the ICR group and 73.5° ± 6.1° in the TMD group, and the difference was statistically significant (*p* < 0.001). Regarding the morphological features of the condylar neck, 7 of 12 patients with ICR exhibited posterior inclination, while 4 of 12 had normal inclination (Table 6). In the TMD group, 6 of 10 patients showed anterior inclination, while 4 patients had a normal inclination.

## 4. Discussion

The higher prevalence of ICR among young women and the identification of estrogen receptors in the TMJ make it highly plausible that hormonal disturbances may be involved in the pathogenesis of ICR [7,8]. Moreover, several clinical studies have reported abnormally low serum 17β-estradiol (E2) levels in women with ICR [16,17]. Furthermore, a comprehensive survey study showed that 10% of patients with ICR had a history of abnormal menstruation, and nearly half had taken contraceptive pills, which can affect the menstrual cycle [18]. Interestingly, E2 concentrations were significantly higher in synovial fluid specimens from the TMJ of patients with ICR than in samples from patients with disc displacement and/or TMJ-OA [19]. Our clinical survey showed that 10 of 12 patients with ICR were women and 1 of 10 female patients with ICR had a medical history of oral contraceptive administration for the treatment of endometriosis and painful menstruation. Thus, sex-hormone imbalances might be an intrinsic factor associated with the onset and progression of ICR. Yuan et al. [13] reported that reduced serum E2 did not contribute to ICR onset, whereas systemic testosterone distributions were found to be related to ICR. Considering that teenage boys rarely exhibit ICR, this hypothesis might be more realistic.

Recently, several researchers have suggested that degenerative diseases of the TMJ, such as ICR and TMJ-OA, are caused by multiple etiological factors, including sex-hormone imbalances and excessive mechanical stress [20,21]. Ootake et al. [21] investigated the interaction between excessive mechanical stress and decreased sex hormones on degenerative changes in the TMJ among orchiectomized (ORX) or ovariectomized (OVX) experimental mice, and demonstrated that excessive mechanical stress to the TMJ aggravates condylar deterioration in OVX and ORX mice. Wu et al. [20] also reported that OVX mice with imbalanced occlusion (increased occlusal height) exhibited the most severe destruction of the condyle, suggesting that the combination of estrogen deficiency and excessive mechanical stress synergistically aggravates condylar resorption. In contrast, OVX and ORX mice without overloads to the TMJ did not exhibit condylar deterioration [20,21]. Although the pathophysiology of ICR remains unclear, intrinsic factors such as hormonal imbalances and immune disturbance might be contributory factors, and extrinsic factors such as mechanical overloading caused by orthodontic treatment, orthognathic surgery, imbalanced occlusion, and macrotrauma might accelerate condylar resorption. The coexistence of intrinsic and extrinsic factors may be essential for ICR onset.

In the present study, all patients with ICR had skeletal Class II malocclusion with an anterior open bite. In particular, the anterior teeth exhibited wear, indicating an occlusal contact between the maxillary and mandibular anterior teeth before ICR development. Based on the interviews of patients with ICR, the occlusion before the development of ICR showed large variation: two patients had an anterior crossbite, two showed crowding, and the remaining had maxillary protrusion. This implies that patients with ICR develop skeletal Class II malocclusion with an anterior open bite due to the reduction in mandibular ramus height caused by condylar resorption. Therefore, the malocclusion might not be an intrinsic factor associated with the onset of ICR or TMDs. However, orthodontic treatment is considered to be associated with the onset and progression of ICR [6,7]. Our results also showed that one-third of patients with ICR had undergone orthodontic treatment before or during the onset of ICR. These findings indicate that orthodontic treatment may lead to excessive or abnormal mechanical force on the TMJ and may be an extrinsic factor associated with the onset and progression of ICR, although such mechanical force on the TMJ is generally negligible under normal conditions.

Arnett et al. [6,7] have described the relationship between TMJ internal derangement and condylar resorption and suggested that TMJ internal derangement may cause condylar resorption, thereby negatively affecting mandibular growth. Furthermore, condylar resorption after surgical mandibular advancement may be associated with coexisting disc displacement [22,23]. Recently, Dalewski et al. [24,25] have explored the genetic susceptibility to anterior disc displacement without reduction (ADDwoR) in European Caucasians and reported that the COL5A1 and ESR1 receptors are a risk factor for ADDwoR, indicating the possibility of genetic susceptibility to ICR. Advancing the mandible in a patient with anteriorly displaced discs causes the discs to remain displaced, as the condyles move toward the superoposterior position in the glenoid fossa due to postsurgical soft tissue tension [23]. This leads to the initiation or worsening of condylar resorption after mandibular advancement. However, patients with mandibular deficiency do not always exhibit anterior disc displacement, and those with normal mandibular growth commonly have disc displacement. Politis et al. [26] reported that the frequency of ICR after mandibular advancement was ≤1%, and that most patients undergoing orthognathic surgery can be managed conservatively. In our survey, a self-reported history of mouth opening disturbance was present in 6 out of 12 patients with ICR, while 3 of 12 patients (25%) had no signs or symptoms of TMD. Kristensen et al. [15] also demonstrated that 12% of patients with ICR had no TMD signs or symptoms. Therefore, TMJ internal derangement with disc displacement is regarded as an intrinsic factor, while surgical mandibular advancement contributes to the onset of ICR is an extrinsic factor. Surgical mandibular advancement might be a factor contributing to ICR development, but it is not sufficient to cause ICR.

The CT findings in our study showed that patients with ICR had significantly different condylar sizes and shapes. This is consistent with the findings in a previous study [15], and the changes in the condylar shape and reductions in the condylar width and height might be due to ICR development. A smaller condylar axial angle may be a specific characteristic of ICR. We studied the stress distribution in the TMJ during clenching using a 3D finite element model and demonstrated that during clenching, the lateral and anterior areas of the condyle received considerably larger compressive stress than the posterior and medial areas [27]. In addition, the lateral and anterior areas of the condyle are likely to experience larger compressive stress during mouth opening [28]. This indicates that bone remodeling in the lateral and medial areas of the condyle during the progression of ICR may be different, resulting in a reduction in the condylar axial angle. Kristensen et al. [15] inferred from the mandibular anatomy that a change in the condylar axial angle may result from condylar resorption from the superior direction because the condylar head is rotated superolaterally after emerging from the ramus. Thus, normal age- and sex-appropriate values and variations of the condylar axial angle are needed for the diagnosis and prognosis of ICR. Furthermore, CT evaluation must be performed in growing patients suspected of having ICR because the changes in condylar shape and reduction in condylar size may precede clinical findings. This may lead to early detection and early stage treatment of ICR.

Our survey-based study has a number of limitations. With respect to the present results, the following points should be noted: First, the participants of the present survey were recruited from a group of orthodontic patients only, whereas previous studies included participants from the orthodontics and/or oral and maxillofacial surgery departments [13,15,29]. This implies that our study may have yielded different results had we recruited participants from the general population. Second, we did not perform MRI examination for patients with ICR, preventing confirmation of anteriorly displaced disc in the TMJ. MRI findings can provide detailed information on the disc position and structure and enable the validation of the correlation between the ICR and ADDwoR, resulting in a more robust research grounding. Finally, the study had a small sample size. To confirm the validity of the results of the present study, in terms of the condylar size and shape, power analysis was performed. According to the results of the power analysis, the required sample sizes were approximately 5.3, 493.2, 4.3, and 4.1 persons for condylar width, length, height, and axial angle, respectively. This indicates that the sample size of the present study was appropriate for condylar width, height, and axial angle, but insufficient for condylar length. Further survey and investigation involving a larger sample size are needed to determine the specific condylar features in patients with ICR.

## 5. Conclusions

In the present study, we explored the intrinsic and extrinsic factors associated with ICR onset and progression in orthodontic patients. The coexistence of intrinsic and extrinsic factors such as sex-hormone imbalance and a history of orthodontic treatment might contribute to the onset of ICR. We suggest that growing patients suspected of having ICR undergo CT evaluation because CT findings may precede clinical findings.

## Figures and Tables

**Figure 1 jcm-11-01552-f001:**
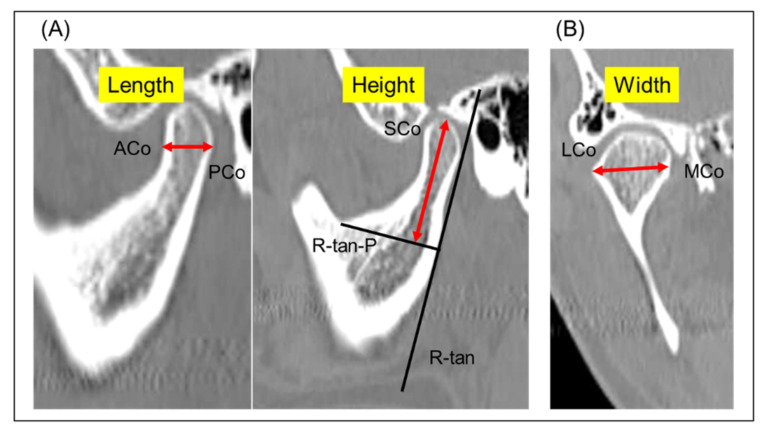
Three-dimensional CT images and the definitions of the measurement items. (**A**) Sagittal view showing condylar length measurement between ACo and PCo, and condylar height as distance from SCo to R-tan-P. (**B**) Coronal view showing condylar width measurement between LCo and MCo. ACo, anterior-most condylar point; PCo, posterior-most condylar point; SCo, superior mandibular condyle; R-tan, a tangent to the posterior surface of the ramus at the deepest point of the mandibular incisura; R-tan-P, a line perpendicular to R-tan; LCo, lateral condylar point; MCo, medial condylar point.

**Figure 2 jcm-11-01552-f002:**
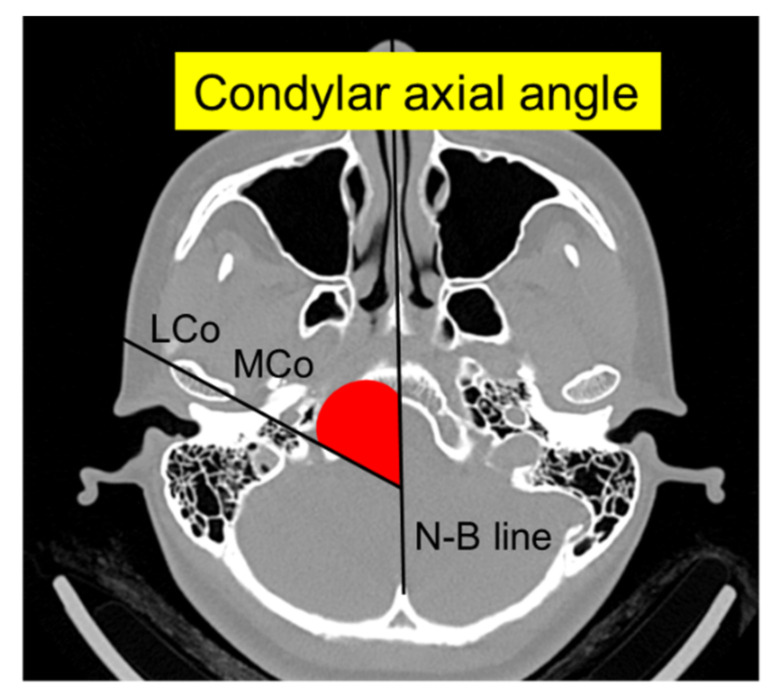
Axial 3D CT image showing the condylar axial angle between the condylar axis (LCo-MCo) and the midsagittal reference line (N–B). LCo, lateral condylar point; MCo, medial condylar point; N–B, Nasion–Basion.

**Table 1 jcm-11-01552-t001:** Prevalence of TMDs and ICR.

	ICR		TMD	
Male	2	0.3	138	20.9
Female	10	0.9	312	29.1
Total	12	0.7	450	25.9

**Table 2 jcm-11-01552-t002:** Prevalence of malocclusion in orthodontic patients with ICR and TMDs.

	ICR (*n*, 12)		TMD (*n*, 450)		Control (*n*, 1273)	
	*n*	%	*n*	%	*n*	%
Maxillary protrusion	12	100.0	122	27.1	256	20.1
Mandibular protrusion	0	0.0	66	14.7	258	20.3
Posterior crossbite	2	16.7	37	8.2	75	5.9
Open bite	9	75.0	52	11.6	55	4.3
Deep bite	0	0.0	9	2.0	23	1.8
Crowding	0	0.0	144	32.0	540	42.4
Other	0	0.0	20	4.4	66	5.2

**Table 3 jcm-11-01552-t003:** Demographic characteristics of the study participants.

	ICR (*n* = 12)	TMD (*n* = 450)	*p* Value
Age (years), mean (SD)	16.4 (4.5)	17.9 (5.1)	
Sex, *n*			
Female	10	312	
Male	2	138	
Reported history, *n* (%)			
Joint sounds	6 (50.0)	203 (45.1)	0.36
Joint pain	5 (41.7)	110 (24.4)	0.17
Difficulty of mouth opening	6 (50.0)	71 (15.8)	<0.01
Headache	2 (16.7)	106 (23.6)	0.58
Stiff shoulder	1 (8.3)	103 (22.9)	0.23
Masticatory muscle pain	2 (16.7)	27 (6.0)	0.13
arafunctional habits	8 (66.7)	292 (64.9)	0.90
Maxillofacial trauma	2 (16.7)	73 (16.2)	0.96
Orthodontic treatment	4 (33.3)	44 (9.8)	<0.01
Objective findings			
Joint sounds	9 (75.0)	261 (58.0)	0.24
Joint pain	3 (25.0)	41 (9.1)	0.06
Masticatory muscle tenderness	1 (8.3)	34 (7.6)	0.92
Mandibular shift at max. opening	4 (33.3)	73 (16.2)	0.11
Maximal mouth-opening (mm)	35.1 ± 6.6	42.4 ± 7.4	<0.01

**Table 4 jcm-11-01552-t004:** Summary of 12 patients with ICR: sex, type of malocclusion, TMD symptoms, and reported history.

	Case 1	Case 2	Case 3	Case 4	Case 5	Case 6	Case 7	Case 8	Case 9	Case 10	Case 11	Case 12
*Sex*	Female	Male	Female	Female	Female	Female	Female	Female	Female	Female	Female	Male
*Age at onset, years*	16	15	12	17	13	14	10	16	14	14	12	5
*Type of malocclusion*	Maxillary protrusion and open bite	Maxillary protrusion and open bite	Maxillary protrusion and open bite	Maxillary protrusion & open bite and posterior crossbite	Maxillary protrusion and open bite	Maxillary protrusion and open bite	Maxillary protrusion and open bite and posterior crossbite	Maxillary protrusion and open bite	Maxillary protrusion and open bite	Maxillary protrusion	Maxillary protrusion	Maxillary protrusion
*TMD symptoms*	(+)	(±)	(−)	(+)	(+)	(−)	(+)	(+)	(+)	(+)	(+)	(+)
*Orthodontic treatment*	(−)	(+)	(+)	(+)	(−)	(−)	(−)	(−)	(−)	(+)	(−)	(−)
*Third molars extraction*	(−)	(−)	(−)	(−)	(−)	(−)	(−)	(−)	(−)	(−)	(−)	(−)
*History of trauma*	(+)	(−)	(+)	(−)	(−)	(−)	(−)	(−)	(−)	(−)	(−)	(−)
*Parafunctional habits*	(+)	(−)	(−)	(+)	(−)	(−)	(+)	(+)	(+)	(+)	(+)	(+)
	Nail biting and chin rest			Thumb sucking and chin rest			Mouth breathing and tongue thrusting	Thumb sucking and mouth breathing	Mouth breathing and chin rest	Chin rest	Tongue thrusting	Nail biting
*Systemic diseases*	(−)	(−)	(−)	(±)	(−)	(−)	(−)	(−)	(+)	(+)	(+)	(−)
*Oral contraceptive use*	(−)	(−)	(−)	(−)	(−)	(−)	(−)	(−)	(−)	(−)	(+)	(−)
*Skeletal pattern (before and after onset)*	Cl II → Cl II	Cl III → Cl II	Cl II → Cl II	Cl III→ Cl I	Cl II → Cl II	Cl II → Cl II	Cl II→ Cl II	Cl II → Cl II	Cl II → Cl II	Cl II → Cl II	Cl II → Cl II	Cl I → Cl I

**Table 5 jcm-11-01552-t005:** Measurements of condylar width, length, and height, and axial angle in the ICR and TMD groups.

	Mean	SD	*p* Value
Condylar width (mm)			
ICR	14.1	2.0	
TMD	18.0	1.6	
Difference	3.9		<0.001
Condylar length (mm)			
ICR	8.0	2.8	
TMD	8.5	1.2	
Difference	0.5		0.51
Condylar height (mm)			
ICR	12.6	4.1	
TMD	21.7	3.5	
Difference	9.1		<0.001
Condylar axial angle (°)			
ICR	44.6	12.1	
TMD	73.5	6.1	
Difference	28.9		<0.001

**Table 6 jcm-11-01552-t006:** Condylar neck inclination in the ICR and TMD groups.

*n* (%)	Anterior	Normal	Posterior
ICR	1 (8.3)	4 (33.3)	7 (58.3)
TMD	6 (60.0)	4 (40.0)	0 (0.0)

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
