# Peer review of "Signs, Symptoms, and Morphological Features of Idiopathic Condylar Resorption in Orthodontic Patients: A Survey-Based Study"

_jcm, 2022, doi:10.3390/jcm11061552_

Round 1

Reviewer 1 Report

Congratulations to the author's team for such an extensive data-based study, the following correction is recommended .

Abstract;

  1. simplify... the terms like "clinico-statistical survey"
  2. title...epidemiological survey...we can say... a survey-based study or a suitable simple term.
  3. line number 17 and 18..." and ICR may result in changes in the condylar shape and reductions in the condylar width and height" the authors must be sure about the findings of the study.
  4. mention research gap related to the topic in the first few lines of abstract...to support "why" the research has been done.
  5.  keywords, MeSH words are encouraged to be included, in alphabetical order.
  6. The introduction section, mention the "justification" of this study in the last paragraph.
  7. Discussion section, add limitation of the study, strengths of the research, and future directions for further research in the field.
  8. overall check the paper for grammatical errors and typos.

Author Response

Reviewer #1:

Congratulations to the author’s team for such an extensive data-based study, the following correction is recommended.

Abstract

  1. Simplify… the term like “clinic-statistical survey”

Response: According to your suggestion, we have changed the term “clinic-statistical survey” to “survey-based study”. (revision: Page 1, line 13)

  1. Title…epidemiological survey…we can say… a survey-based study or a suitable simple term.

Response: We have changed the title into “Signs, Symptoms, and Morphological Features of Idiopathic Condylar Resorption in Orthodontic Patients: A Survey-Based Study”. (revision: Page 1, lines 2-4)

  1. Line number 17 and 18…” and ICR may result in changes in the condylar shape and reductions in the condylar width and height” the authors must be sure about the findings of the study.

Response: We agree with your point. This sentence did not reflect the result obtained from the current data. Accordingly, we have deleted this sentence. (revision: Page 1, line 23)

  1. Mention research gap related to the topic in the first few lines of abstract… to support “why” the research has been done.

Response: According to your recommendation, we have revised the first few lines of the Abstract as follows:

… However, no specific cause of ICR has been identified. To explore the specific causes of the onset and progression of ICR, we performed a survey-based study on ICR in orthodontic patients and described its subjective symptoms, clinical signs, and condylar morphological features. (revision: Page 1, lines 12-15)

  1. Keywords, MeSH words are encouraged to be included, in alphabetical order.

Response: We deleted and added some keywords as follows:

Keywords: idiopathic condylar resorption; orthodontic patients; orthognathic surgery; progressive condylar resorption; temporomandibular disorders

  1. The introduction section, mention the “justification” of this study in the last paragraph.

Response: Similar to the Abstract, the last paragraph of the Introduction section was revised. (revision: Page 2, lines 61-62)

  1. Discussion section, add limitation of the study, strengths of the research, and future directions for future research in the field.

Response: Thank you for your insightful comment. We added one paragraph about the study limitations, including the small sample size. (revision: Page 8, lines 376-392)

  1. Overall check the paper for grammatical errors and typos.

Response: The initial manuscript submitted was checked and revised by an English editing service “Editage”, and we had the manuscript checked and revised for the second time to ensure its quality. (several revisions)

Reviewer 2 Report

This is an interesting article trying to connect dots in-betweensigns, symptoms, and morphological features of idiopathic condylar resorption in orthodontic patients; topic is very important, however disease is rare, yet many concomittant conditions and factors are commonly overlooked in orofacial pain dx and treatment not only prior to ortho: pain perception, pain referral, occlusal factors and general health conditions in orofacial pain management. Therefore I feel that this is the most important paper I have revieved in some time. Also evaluation of TMD patients with myofascial pain and/or referred pain is still very undervalued topic, so I readed an entire manuscript carefully and with a great interest. The studied sample is very small, however I acknowledge that the disease is rare and authors have had spent probably a lot of time to acquire it from the database. Overall data collected and study execution quality are good, unfortunately an introduction section should be slightly rewritten for clarity, as it seems that some major points are missing here, see below - some criticism should be raised prior to publication:

Abstract:

This section should be structurized according to typical MDPI policy and exact science - it reads like and introduction setion in humanities now - see https://www.mdpi.com/journal/jcm/instructions

Also adding few more keywords should improve availability through search engines and would eventually lead to improved citation odds

Introduction

The very important new paper by Mercuri et. al https://pubmed.ncbi.nlm.nih.gov/31685348/ is missing in ICR description. There are actually also two up-to-date articles by Dalewski et al. scrutinizing role of ADDwoR, ESR1 and COL5A1 receptors role as one of many factors which might are contributing to JCR, hence please incorporate and cite at least https://pubmed.ncbi.nlm.nih.gov/33101543/ and https://pubmed.ncbi.nlm.nih.gov/34572072/ as no genetic susceptibilty to ICR was mentioned in this section, TMJ disc displacements without reduction underwent at the young age are not mentioned here either.

Materials and methods

L109-111 - 'oral parafunctions' term is obsolete and not used anymore, look for replacement according to ICSD-2 with latter changes - please rewrite this paragraphs with relevant citation(s)

Results

As I mentioned before - ICR is more common in young patients (F>M) who underwent DDwoR at early age without successful treatment or disc-recapture - without prior orthognatic surgery - these cases are strictly connected with limited mouth opening, which authors also already emphasized. Hence in ref. 21 and 22 there is only post-surgical disc displacement mentioned; I wish authors had evaluated this, yet they did not. Please try include such data next time writing ICR article - you have a very potent patient database.

Discussion

Papers I brought up earlier might be used in this section either.

Conclusions

There should be a brief paragraph here about study limitatons - e.g. study sample size

Good luck!

Author Response

Reviewer #2

This is an interesting article trying to connect dots in-between signs, symptoms, and morphological features of idiopathic condylar resorption in orthodontic patients; topic is very important, however disease is rare, yet many concomitant conditions and factors are commonly overlooked in orofacial pain dx and treatment not only prior to ortho: pain perception, pain referral, occlusal factors and general health conditions in orofacial pain management. Therefore I feel that this is the most important paper I have received in some time. Also evaluation of TMD patients with myofascial pain and/or referred pain is still very undervalued topic, so I read an entire manuscript carefully and with a great interest. The studied sample is very small, however I acknowledge that the disease is rare and authors have had spent probably a lot of time to acquire it from the database. Overall data collected and study execution quality are good, unfortunately an introduction section should be slightly rewritten for clarity, as it seems that some major points are missing here, see below-some criticism should be raised prior to publication:

Response: We thank you for your valued comments.

Abstract:

This section should be structurized according to typical MDPI policy and exact science – it reads like and introduction section in humanities now – see https://www.mdpi.com/journal/jcm/instructions.

 Response: According to your suggestion, we have included a structured Abstract. (revision: Page 1, lines 11-26)

Also adding few more keywords should improve availability through search engines and would eventually lead to improved citation odds.

Response: We have deleted and added some keywords as follows:

Keywords: idiopathic condylar resorption; orthodontic patients; orthognathic surgery; progressive condylar resorption; temporomandibular disorders

Introduction

The very important new paper by Mercuri et al. https://pubmed.ncbi.nlm.nih.gov/31685348/ is missing in ICR description. There are actually also two up-to-date articles by Dalewski et al. scrutinizing role of ADDwoR, ESR1 and COL5A1 receptors role as one of many factors which might are contributing to ICR, hence please incorporate and cite at least https://pubmed.ncbi.nlm.nih.gov/33101543/ and https://pubmed.ncbi.nlm.nih.gov/34572072/ as no genetic susceptibility to ICR was mentioned in this section, TMJ disc displacements without reduction underwent at the young age are not mentioned here either.

Response: Thank you for your insightful suggestion. We have cited the three papers you recommended. The paper by Mercuri et al. was cited in the Introduction section and the remaining two papers by Dalewski et al. were cited in the Discussion section. (revisions: Page 1, lines 41-43 and Page 7, lines 338-341)

Materials and methods

L109-111 – ‘oral parafunctions” term is obsolete and not used anymore, look for replacement according to ICSD-2 with latter changes – please rewrite this paragraph with relevant citation(s).

Response: We were confused because the term ‘oral parafunctions’ has been commonly used in scientific journals including JCM (https://mdpi.com/2077-0383/11/3/589/). However, based on your recommendation, we changed it to ‘parafunctional habits’. (revisions: Page 3, line 117 and Page 5, line 223)

Results

As I mentioned before – ICR is more common in young patients (F>M) who underwent DDwoR at early age without prior orthognathic surgery – these cases are strictly connected with limited mouth opening, which authors also already emphasized. Hence in ref. 21 and 22 there is only post-surgical disc displacement mentioned; I wish authors had evaluated this, yet they did not. Please try include such data next time writing ICR article – you have a very potent patient database.

Response: Thank you for the excellent suggestion. We will try to include this information in future work on ICR. (no revision)

Discussion

Papers I brought up earlier might be used in this section either.

Response: We have added several sentences in the fourth paragraph of the Discussion section and cited the two papers you recommended. (revision: Page 7, lines 338-341)

Conclusions

There should be a brief paragraph here about study limitations – e.g. study sample size.

Response: We thank you for your helpful recommendation. We have added one more paragraph on the limitations of this study in the Discussion section. (revision: Page 8, lines 376-392)

Round 2

Reviewer 2 Report

All of my remarks were successfully addressed